

# Cost-benefit trade-offs of bird activity in apple orchards

Rebecca K. Peisley[1,2], Manu E. Saunders[2] and Gary W. Luck[2]

[1] School of Environmental Sciences, Charles Sturt University, Albury, New South Wales, Australia
[2] Institute for Land, Water and Society, Charles Sturt University, Albury, New South Wales, Australia

## ABSTRACT

Birds active in apple orchards in south–eastern Australia can contribute positively (e.g., control crop pests) or negatively (e.g., crop damage) to crop yields. Our study is the first to identify net outcomes of these activities, using six apple orchards, varying in management intensity, in south–eastern Australia as a study system. We also conducted a predation experiment using real and artificial codling moth (*Cydia pomonella*) larvae (a major pest in apple crops). We found that: (1) excluding birds from branches of apple trees resulted in an average of 12.8% more apples damaged by insects; (2) bird damage to apples was low (1.9% of apples); and (3) when trading off the potential benefits (biological control) with costs (bird damage to apples), birds provided an overall net benefit to orchard growers. We found that predation of real codling moth larvae was higher than for plasticine larvae, suggesting that plasticine prey models are not useful for inferring actual predation levels. Our study shows how complex ecological interactions between birds and invertebrates affect crop yield in apples, and provides practical strategies for improving the sustainability of orchard systems.

## INTRODUCTION

Wild animals in agroecosystems interact with crops in complex ways (e.g., direct consumption, pollination, biological control of crop pests, and nutrient cycling) that may reduce, increase or have a benign effect on crop yield (*Borkhataria et al.*, *2012*; *Klosterman et al.*, *2013*; *Klatt et al.*, *2014*). These effects can have a substantial impact on annual production, resulting in significant increases or declines in yield quantities and market values (*Losey & Vaughan*, *2006*; *Murray, Clarke & Ronning*, *2013*). When considering these benefits and costs together, it is clear that trade-offs exist. For example, the beneficial activity of insectivorous birds preying on pest insects in an orchard and reducing insect damage to fruit is traded off against the detrimental activity of the same birds preying on beneficial pollinators resulting in reduced fruit set. Examining the complexities of animal activity within agroecosystems can highlight these trade-offs and allow for calculation of the net outcome (benefits minus costs) of animal activities on production. This has been considerably overlooked in the literature, with very few studies looking at both costs and benefits of animal activity in the same context (*Luck*, *2013*; *Peisley, Saunders & Luck*, *2015*;

Corresponding author
Rebecca K. Peisley,
rpeisley@csu.edu.au

*Saunders et al.*, *2016*). Birds are commonly found in almost all agroecosystems and their foraging activity often results in significant beneficial or detrimental effects on crop yields, making them an excellent case study of cost-benefit trade-offs of animal activity in crops.

Birds may significantly increase crop yields by predating on pest invertebrates (*Mols & Visser*, *2002*; *Johnson, Kellermann & Stercho*, *2010*; *Karp et al.*, *2013*; *Maas, Clough & Tscharntke*, *2013*; *Ndang'ang'a, Njoroge & Vickery*, *2013*). For example, a study in the Blue Mountains of Jamaica by *Kellermann et al.* (*2008*) found that birds reduced coffee berry borer (*Hypothenemus hampei*) damage by up to 14%, by consuming this major insect pest, which increased the coffee (*Coffea spp.*) crop market value by as much as US$105/ha. Similarly, *Mols & Visser* (*2007*) found that great tits (*Parus major*) reduced caterpillar damage to Dutch apple (*Malus domestica*) orchards by up to 50% compared to orchards without the bird species.

However, many bird species, including parrots (*Bomford & Sinclair*, *2002*) and passerines (*Kross, Tylianakis & Nelson*, *2012*), can inflict costs to growers by consuming crops. For example, European blackbirds (*Turdus merula*) and common starlings (*Sturnus vulgaris*) can cause severe damage to grape (*Vitis spp.*) and blueberry (*Vaccinium spp.*) crops (*Avery et al.*, *1996*; *Somers & Morris*, *2002*; *Kross, Tylianakis & Nelson*, *2012*). Birds can also indirectly impact on crop yields by consuming beneficial insects such as pollinators or natural enemies (e.g., *Galeotti & Inglisa*, *2001*).

Previous research found that apple orchards in south–eastern Australia contain a suite of different bird species with the potential to inflict costs and/or provide benefits to fruit production (*Luck, Hunt & Carter*, *2015*). Birds can benefit crop yields by consuming apple pests (e.g., codling moth (*Cydia pomonella*)) and removing unwanted fruit after harvest, thereby reducing disease risk; however, they can also consume and damage fruit before harvest or prey on insects beneficial to apple production (e.g., pollinators). The net outcome of these activities has not yet been considered.

Therefore, the aims of our study were as follows: (1) to determine if excluding birds from branches of apple trees (via netting) in south–eastern Australia resulted in greater insect damage to fruit (indicating that birds may contribute to controlling insect pests) and reduced crop yields; (2) to determine if bird damage to apples on open branches was higher than on netted branches, reducing crop yields; and (3) calculate a net outcome of bird activity, trading off the potential benefits of birds controlling insect pests vs. birds directly consuming fruit. We also conducted an experiment using real and artificial codling moth larvae to gain further insight into what bird species (or insect predators) might be preying on pest invertebrates in the orchards.

## MATERIALS AND METHODS

### Animal ethics

This research was conducted with approval from the Charles Sturt University Animal Care and Ethics Committee (approval number 14/040).

**Table 1  Orchards were classified along a gradient of intensity based on factors that are known to influence bird communities.** Distance of the study area to unmanaged vegetation was considered the most important intensity factor, followed by distance of the study area to farm buildings, and whether pesticide sprays were used.

| Intensity ranking (lowest to highest) | Orchard | Minimum distance to unmanaged vegetation (m) | Amount of closest continuous unmanaged vegetation (ha) | Minimum distance to farm buildings (m) | Pesticide spray used |
|---|---|---|---|---|---|
| 1 | Orchard 1 | 0 | >40 | ~100 | No |
| 2 | Orchard 2 | <10 | <1 | ~550 | No |
| 3 | Orchard 3 | 0 | >6 | ~100 | Yes |
| 4 | Orchard 4 | ~5 | <2 | ~200 | Yes |
| 5 | Orchard 5 | ~50 | >15 | ~10 | Yes |
| 6 | Orchard 6 | >280 | <0.5 | ~250 | Yes |

## Study sites

Our study was conducted across six apple orchards in three major apple growing regions in Australia: Batlow, southern NSW (average annual rainfall 1283.0 mm, average annual temperature 6.0 °C–16.9 °C (*Australian Bureau of Meteorology*, *2015*)), Shepparton, central Victoria (average annual rainfall 506.4 mm, average annual temperature 8.4 °C–22.6 °C degrees (*Australian Bureau of Meteorology*, *2015*)) and Harcourt, central Victoria (average annual rainfall 696.9 mm, average annual temperature 7.7 °C–19.8 °C (*Australian Bureau of Meteorology*, *2015*)). All orchards differed in their management practices and landscape composition. Therefore, rather than focusing on categorical comparisons that can overlook ecological complexity (e.g., organic vs. conventional; *Winqvist, Ahnström & Bengtsson*, *2012*), we ranked orchards along a gradient of intensity based on factors that are known to influence bird communities in agricultural landscapes (*Bennett & Ford*, *1997*; *Benton et al.*, *2002*; *Tscharntke et al.*, *2008*; *Luck, Triplett & Spooner*, *2013*). These included (listed in order of importance) proximity of the orchard to unmanaged natural or semi-natural vegetation, the amount of closest continuous unmanaged vegetation, proximity to large farm buildings, and if the orchard used pesticide sprays (Table 1). For example, native vegetation is known to be important for the presence of birds in agroecosystems (*Bennett & Ford*, *1997*; *Tscharntke et al.*, *2008*); *Luck, Triplett & Spooner* (*2013*) found that farm buildings were negatively associated with bird abundance in almond orchards (likely owing to the frequency of human activity around buildings); and farms that use pesticides can have lower invertebrate numbers which could in turn reduce the food source available to birds (*Benton et al.*, *2002*).

Apple varieties differed across the orchards. These included Pink Lady (Orchard 1, Orchard 4, Orchard 5, Orchard 6), Royal Gala (Orchard 1, Orchard 3), Granny Smith (Orchard 1, Orchard 2, Orchard 4), Golden Delicious (Orchard 1, Orchard 2), Cox (Orchard 1, Orchard 4), Gravenstein (Orchard 1) and Sundowner (Orchard 4). Tree age also varied across orchards (2–20 years), however all trees were established and producing fruit.

## Focal study trees

Within these orchards, sixty apple trees (ten trees per orchard) were systematically selected to be monitored across the entire season for bird and insect damage. Each tree was spaced at

least 15 m apart; this distance was determined by the size of the smallest orchard (Orchard 1, which was approximately 1 ha). Half of the trees were located near orchard edges adjacent to unmanaged vegetation (established woodland at all orchards except Orchard 6, where an unmanaged grassy meadow was the most proximate area of non-crop vegetation), as this was expected to be a potential source of birds (*Tscharntke et al.*, *2008*). The other five trees were located in the centre of the orchard to facilitate a comparison of bird activity (and subsequent costs and benefits) adjacent to and more distant from unmanaged vegetation.

Before flowering (August 2014), two branches of similar height on each tree were selected and marked with flagging tape. One branch was left 'open' under natural conditions and the other was enclosed with white diamond mesh bird netting (15 mm mesh). These 'netted' branches allowed insects to access flowers and fruit while excluding birds. Treatments were paired on each tree to control for differences between trees. The height of the lowest point of the branch, and the distance of each tree to the orchard edge and the nearest patch of unmanaged vegetation was recorded.

## Bird surveys

RKP conducted bird surveys between sunrise and 11.00 am at four key times of the growing season: full bloom (September 2014), early fruit set (October 2014), pre-harvest (December 2014) and post-harvest (May 2015). At each of these times, the species, abundance and behaviour of birds within the apple orchards and adjacent areas of unmanaged vegetation (except at Orchard 6 where there was no adjacent unmanaged vegetation) were identified. This was to determine what bird species occurred in the vicinity of each orchard.

Timed searches with a pre-determined stopping rule were conducted to identify species richness in orchards and unmanaged vegetation (*Miller & Cale*, *2000*; *Watson*, *2003*). An active search method was used whereby the observer searched each orchard or unmanaged vegetation patch for 15 min and recorded every new bird species observed. If a new species was observed within 5 min after the initial 15 min had elapsed, an extra 5 min was added to the search time. This was repeated until no new species were observed in an additional 5 min period. Birds that were flying greater than 5 m above the canopy were excluded, unless they were foraging (e.g., welcome swallows (*Hirundo neoxena*) hawking insects).

To gain a measure of bird abundance, five 'points' were chosen within the orchard (Corner 1, Corner 2, Corner 3, Corner 4 and Centre). Observations were started 1 min after arrival at each point, to allow birds to settle. The identity and abundance of each bird was then recorded for 5 min. A short survey period was employed to reduce the risk of double-counting the same individuals (*Gregory, Gibbons & Donald*, *2004*). The distance to every bird observed was measured to facilitate the calculation of detectability corrected density measures for each species. However, bird abundance (i.e., sample size per species) was not sufficient to allow this calculation, so only general abundance measures are provided in the results.

The 10 focal trees per orchard were also observed for a maximum of 30 min and the identity and activity (particularly foraging behaviour) of every bird that visited was recorded (Supplemental Information 1 and Fig. S1). Birds were classified into the following feeding groups using the published literature to identify the main component of each species' diet:

omnivore, herbivore, insectivore, granivore and carnivore. If a species had two or more main components in its diet (e.g., fruit and insects), then it was classified as an omnivore (*Del Hoyo et al.*, *1992–2013*).

## Apple damage assessments

Immediately before harvest, all apples on both open and netted branches were counted, and the number of any insect or bird damaged apples were recorded. Published damage guides (e.g., *Victorian Department of Natural Resources and Environment*, *2002*; *Victorian Department of Environment and Primary Industries*, *2014*) and personal observations were used to identify damage type. For example, bird damage is often categorised by triangular beak marks or deep gouges (Figs. S2A and S2B) and insect damage usually occurs as circular tunnels or characteristic scar marks (Figs. S2C and S2D). Aborted fruit inside nets were not counted, as we could not ascertain how many fruit had been dropped from comparable open branches. The difference in yield between branches was classified as the percentage of fruit on each branch that had damage.

## Larvae predation

An additional 72 apple trees (12 trees per orchard) were systematically selected to conduct experiments with the primary aims of identifying bird and/or insect predators of pest insects in the orchards, and assessing the validity of using artificial prey analogues to measure predator activity. To our knowledge, no other study has compared predation events on real and artificial prey in the same context at the same time in any crop system (but see *Sam, Remmel & Molleman*, *2015* for an example in tropical forests). Each tree was spaced 15 m apart and was located at orchard edges adjacent to unmanaged vegetation.

Codling moth larvae are a serious insect pest of apples and other pome fruit, with the potential to ruin almost entire fruit crops (*Williams*, *2002*). Their natural occurrence in orchards means that they should be an easily recognizable food source for insectivorous birds which prey on invertebrates on the surface of the fruit, foliage or branches of apple trees. This made them an ideal 'model pest' to use to assess which bird species may potentially be providing biological control in apple orchards. Codling moths emerge from over-wintering under the bark of trees in early spring and lay eggs which hatch and infect young fruit. There can be up to three life-cycles in one apple growing season (*Williams*, *2002*).

We assessed predation on codling moth larvae using artificial and real moth larvae. Artificial larvae were constructed using plasticine. White and pink plasticine was mixed together and rolled into 1.7 mm × 15 mm cylinders to create the creamy pink colour and shape of mature coding moth larvae. A 2 mm diameter ball of black plasticine was also attached to one end as the head (Fig. 1). In total, 360 artificial larvae were constructed for this experiment. Studies have shown that using artificial, plasticine larvae can be an effective alternative to real larvae when conducting predation experiments (*Koh & Menge*, *2006*; *Howe, Lövei & Nachman*, *2009*; *Tvardikova & Novotny*, *2012*). They are easier to source and store, and animal damage can often be identified (e.g., insect, bird or mammal). However, few studies have compared predation rates on real vs. artificial prey in the same
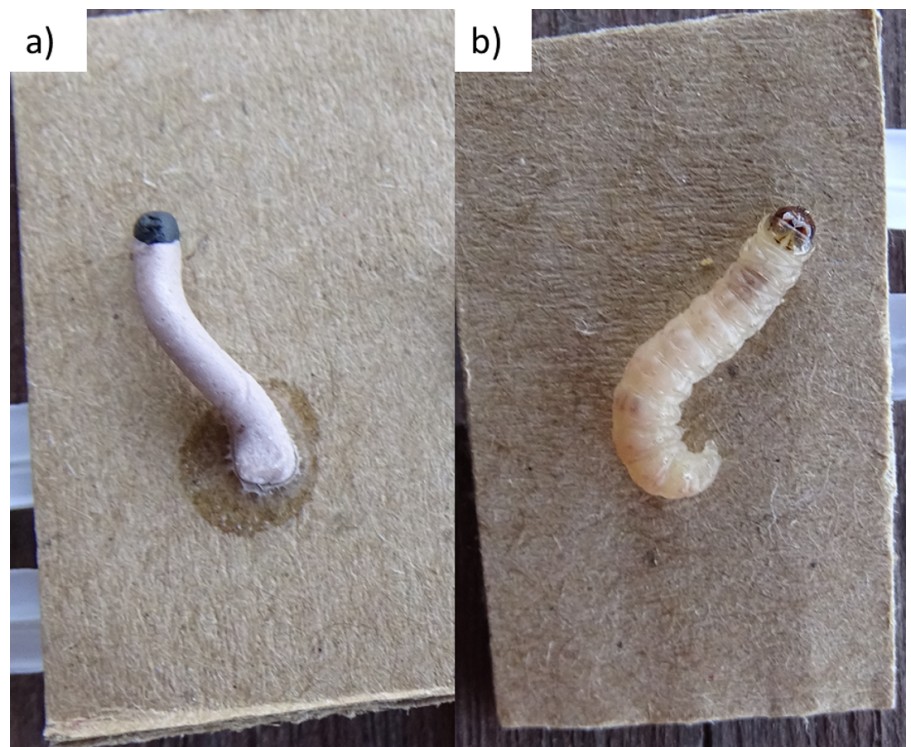

**Figure 1** (A) plasticine larvae, (B) real codling moth larvae.

context to ascertain the usefulness of artificial prey models as analogues of predation rates on real prey (*Sam, Remmel & Molleman*, *2015*).

For real larvae, Agriculture Victoria supplied 360 dormant codling moth larvae (Fig. 1). These were killed in the freezer before use to prevent any live larvae from escaping into orchards.

Half of the artificial and real larvae were stuck individually onto separate 10 mm × 20 mm pieces of cardboard using double sided tape (Fig. 1), and the other half grouped into 36 groups of 10 (five real and five artificial) and stuck together onto 36, 100 mm × 200 mm pieces of cardboard (Fig. S3). Our codling moth larvae likely experienced higher rates of predation than would occur naturally as they were exposed openly on branches and pieces of cardboard; however, our aim here was to determine differences in predation rates between real vs. plasticine baits, rather than infer actual predation rates in orchards.

After fruit set (October 2014, coinciding with the fruit set bird surveys), the larvae were set out in the orchard. The growing season was several weeks earlier than usual due to warm weather and codling moth were active throughout the orchards. We also observed codling moth larvae on apple fruit at this time. Ten individual (five real and five plasticine) codling moth larvae were stuck onto branches of six trees in each orchard. Larvae were stuck on branches of similar height, close to apple clusters and easily visible for potential bird predators to find (Fig. S3). On the remaining six trees, a piece of cardboard containing five real and five artificial larvae was tied to the trunk of the tree or on an adjoining lateral branch (Fig. S3). These cards were monitored by Reconyx HC500 remote motion sensor

cameras to determine which bird species were preying on the larvae. Cameras were set to take a burst of three photos each time they were triggered to enable easy identification of species. Birds most actively forage in the morning so all larvae were set out in the evening to allow for the first morning of bird activity to occur uninterrupted.

### Predation surveys

Twenty-four hours after the larvae were exposed every piece of cardboard was checked and larval status was recorded (i.e., present, removed or damaged). Damaged plasticine larvae were inspected further and damage type was identified as either insect or bird ('other', e.g., mammal damage, was also a category based on guides in the literature (*Howe, Lövei & Nachman*, *2009*); however, only bird and insect damage was encountered). Insect damage was identified with a magnifying glass and occurred as pinpricks and/or pincer marks (Figs. S4A and S4B). Bird damage was identified by straight beak marks (Fig. S4C). Damage assessments were repeated at the same time every day until larvae had been exposed for 5 days. Other studies using artificial caterpillars had exposure times ranging from 24 h to 6 days (*Loiselie & Farji-Brener*, *2002*; *Posa, Sodhi & Koh*, *2007*; *Howe, Lövei & Nachman*, *2009*; *Tvardikova & Novotny*, *2012*).

## STATISTICAL ANALYSIS

### Apple damage

The response variables for the first experiment were the percentage of apples damaged by birds and the percentage of apples damaged by insects on open and netted branches. However, bird damage was very low (average of $1.9\% \pm 4.8$ (95% confidence interval (CI)), and only detected at two sites (Orchard 1 and Orchard 2), so was not included for analysis.

Spearman correlation analysis was used to identify correlated explanatory variables, with only one variable of each correlated pair (correlation defined as $r > 0.3$) being included in models (see 'Results'). A generalized linear model using a Poisson distribution was fitted using the GENMOD procedure in SAS/STAT (SAS Institute, Cary, NC, USA) to determine differences in insect damage to apples between open and netted branches, and whether this difference was influenced by growing Region (Batlow, Shepparton or Harcourt), Orchard (Orchard 1, Orchard 2, Orchard 3, Orchard 4, Orchard 5, or Orchard 6), or Location of the tree in the orchard (edge or interior) (*McCullagh & Nelder*, *1989*). Since every tree was surveyed twice (one open and one netted branch), a block design using generalized estimating equations was used to account for lack of independence in the data resulting from repeat measures (*Liang & Zeger*, *1986*).

### Larvae predation

The response variables for the second experiment were the percentage of plasticine larvae attacked or removed from branches after 5 days, and the percentage of real larvae attacked or removed from branches after 5 days. The explanatory variables were Larvae type (real or plasticine), growing Region (Batlow, Shepparton or Harcourt) and Orchard (Orchard 1, Orchard 2, Orchard 3, Orchard 4, Orchard 5, or Orchard 6). Region was the fixed factor

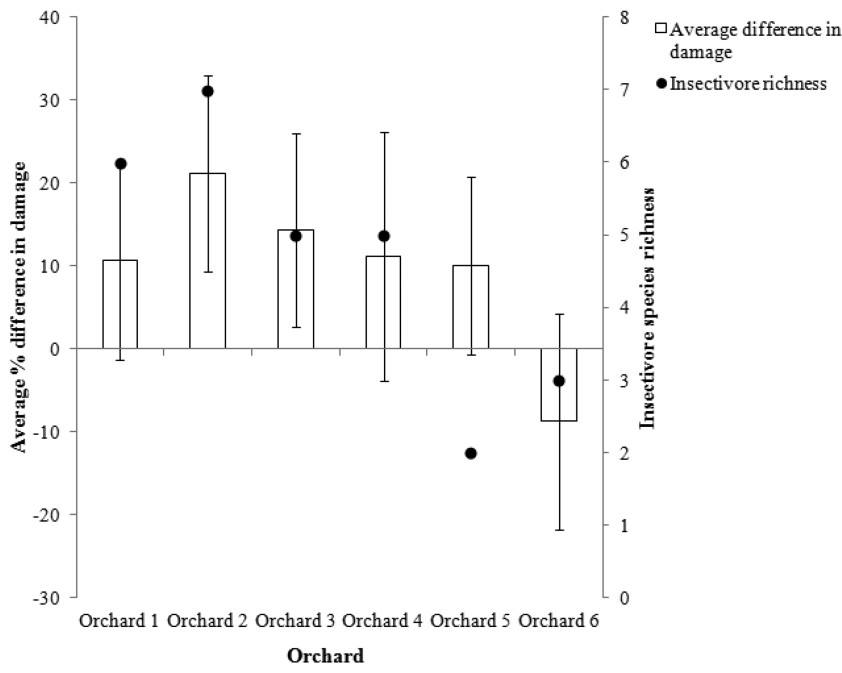

**Figure 2** Average difference in % of insect damaged apples between open and netted branches, compared with insectivorous bird species richness ($n = 120$ branches). All orchards except Orchard 6 received more insect damage on netted (bird excluded) branches (Error bars show 95% CI). Orchards are listed 1–6 from lowest intensity management to highest intensity.

of interest, Orchards were nested within Region, and Larvae type was a repeated measure factor.

A linear mixed model analysis was performed in SAS/STAT (SAS Institute, Cary, NC, USA) fitting the repeated measure, (Larvae type) using an unstructured correlation matrix to determine if: (1) the rate of removal differed between real and artificial larvae; and (2) if there was a relationship between the amount of larvae taken and growing region or orchard.

## RESULTS

### Apple damage

At all sites except Orchard 6 there was more insect damage on netted branches than open branches, with an average of 18.6% (95% CI [13.5–25.5]) of fruit damaged on netted branches and an average of 5.8% (95% CI [3.9–8.7]) of fruit damaged on open branches (least squares mean; $p < 0.005$) (Fig. 2). Therefore, the difference in damage between netted and open branches was positive (12.8%, 95% CI [10.8–17.9]). This suggests that birds were contributing to the biological control of insect pests in most orchards. Orchard 2 had the highest insectivorous bird species richness across the whole season and the greatest average difference in damage between open and netted branches (21.1% ± 11.9 (95% CI)) (Fig. 2). The highest amount of insect damage was also recorded at Orchard 2 with an average of 39.2% (±9.2% (95% CI)) of apples on netted branches damaged (Fig. S5).

The distance of an apple tree to unmanaged vegetation was correlated with Orchard ($r = 0.452$, $p = <0.01$), and did not greatly improve the QIC value for the model. Orchard was considered a more important variable for capturing other aspects of variation across sites so Distance was excluded. The final explanatory variables selected were as follows: the fixed variables Region (Batlow, Shepparton or Harcourt), Orchard (Orchard 1, Orchard 2, Orchard 3, Orchard 4, Orchard 5 or Orchard 6), and Tree Location (Edge or Interior); and the random variable was an Orchard × Region interaction term.

The orchard that the tree was in was the only variable that significantly explained the difference in insect damage between netted and open branches ($p < 0.005$). The difference was greater in the less intensely managed orchards (see ranking scale, Table 1 and Fig. S5), i.e., birds were providing more biological control in these orchards. The region the tree was in (i.e., Shepparton, Harcourt or Batlow) possibly also influenced how much of an effect the exclusion of birds had on insect damage to apples ($p = 0.052$). The greatest difference in damage between netted and open branches occurred in Batlow (average difference of 12.2% ± 7.2 (95% CI)), followed by Harcourt (average difference of 10.9% ± 8.6 (95% CI)), and Shepparton (average difference of 6.2% ± 10.7 (95% CI)).

Insectivorous bird species richness was measured at the orchard level so could not be included in the main analysis, however, when the orchard management intensity was considered against insectivorous bird species richness they were significantly negatively correlated (Spearman $= -0.870$, $p < 0.05$). That is, the least intensively managed orchards had the higher insectivore richness (Fig. S5). The average difference in damage between netted and open branches was greater in orchards that had higher insectivore richness (Spearman $= 0.841$, $p < 0.05$).

## Bird species richness

Overall, 39 different bird species were detected in the six orchards. Thirty-six species were native and three were introduced (Table S1). Thirty-four different species were observed during the full bloom period, 19 species were observed during fruit set, 15 species were observed at harvest and 21 species were observed post-harvest. All species observed in the orchards were also observed in adjacent patches of unmanaged vegetation, as well as an additional eight species (Supplemental Information 1).

Overall, Orchard 3 had the highest species richness (18), followed by Orchard 1 (16), Orchard 2 and Orchard 4 (15 each), Orchard 5 (12) and Orchard 6 (8) (Fig. 3). When species richness was broken down by time of season, Orchard 3 had the highest species richness during the flowering period, with 14 species, followed by Orchard 1 with 12 different species. At fruit set, species richness was also highest at Orchard 3 (9 species), followed by Orchard 2 (7 species) and Orchard 1 (5 species). At harvest time, Orchard 4 had the highest species richness (9 species), while at post-harvest species richness was highest at Orchard 2 (8 species).

## Feeding guilds

Birds were classified by their main feeding type. Omnivores (birds that consumed insects and plant material) were the most common feeding type in all orchards across the entire

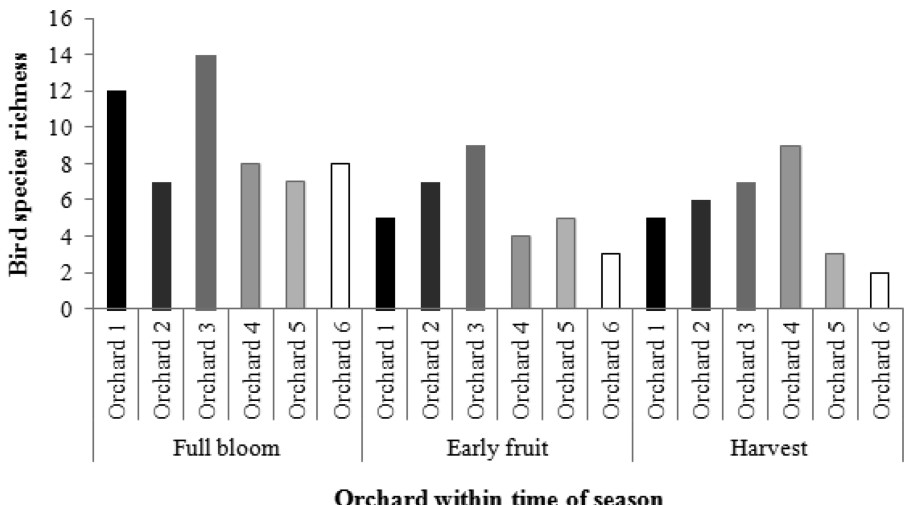

**Figure 3** **Bird species richness in each orchard at key times of the growing season.** Orchards are listed from 1–6 from lowest intensity management to highest intensity.

season, with the exception of Orchard 4 during harvest where there were more insectivorous species. Across the whole season, Orchard 2 had the highest species richness of insectivorous birds (7 species), followed by Orchard 1 (6 species), Orchard 3 and Orchard 4 (5 species each), Orchard 6 (3 species) and Orchard 5 (2 species).

When this was broken down into time of season, four insectivorous bird species were recorded at Orchard 1 and Orchard 3 during flowering, while Orchard 2 and Orchard 6 had three insectivorous species. During early fruiting, only three orchards had predominately insectivorous bird species: Orchard 3 three species, Orchard 2 two species, Orchard 1 one species. At harvest time, Orchard 4 had the highest number of insectivorous species (5) while post-harvest, Orchard 2 had the highest insectivore species richness (5) (Fig. S6).

### Larvae predation experiment

Real larvae were preyed on significantly more than plasticine larvae ($p = <0.0001$; Fig. 4). There was no clear pattern to this: some orchards with high predation rates on real larvae also had high predation rates on plasticine larvae, however this was not always the case. Region also explained the difference in the predation rate ($p = <0.001$), with the most larvae removed in Batlow, followed by Harcourt and then Shepparton (Fig. 4).

Several insectivorous bird species were observed within the orchards during early fruit set and these could potentially be providing biological control of insect pests (Table S1). However, the motion-sensor cameras only detected two birds (a superb fairy-wren (*Malurus cyaneus*) and a satin bowerbird (*Ptilonorhynchus violaceus*)) potentially feeding on the larvae at one orchard (Orchard 3). All predation of plasticine larvae was by insects, with the exception of two bird predated larvae in Orchard 3. The motion-sensor cameras recorded earwigs (*Dermaptera sp.*, including the European earwig (*Forficula auricularia*)) predating upon the larvae in Orchard 1, Orchard 2 and Orchard 5 (Fig. S7). The damage to larvae in the remaining orchards was similar (pincer marks and 'chewed' sections in the plasticine

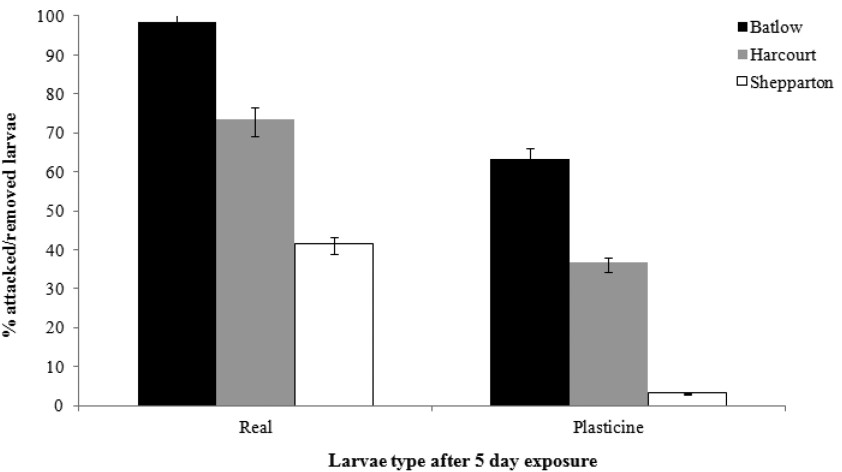

**Figure 4   Least squares means of the number of real and plasticine larvae attacked/removed from each region.** Error bars show 95% confidence intervals.

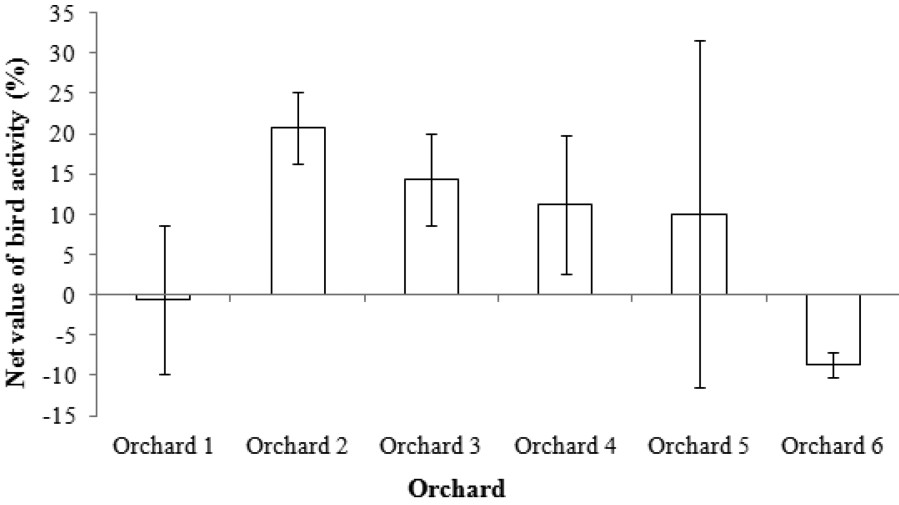

**Figure 5   Net value of bird activity in apple orchards when considering a cost-benefit trade-off (i.e., reduction in insect damaged fruit minus amount of bird damaged fruit).** Orchards are listed 1–6 from lowest intensity management to highest intensity. Error bars show 95% confidence intervals.

larvae, and only the hard head remaining for real larvae) and therefore suggests earwigs or similar predatory invertebrates were also responsible for the majority of predation. Ants (unknown sp.) were also observed eating the real larvae.

## The net value of bird activity

Birds damaged an average of 1.9% of apples within the study orchards, while they reduced the amount of insect damaged apples by an average of 12.8%. This result suggests that birds are providing a net benefit value to orchard growers, reducing damage by an average of 10.9%. When considered at the orchard level, all orchards except two had an overall positive net value of bird activity (Fig. 5).

## DISCUSSION

This study considered the cost-benefit trade-off between birds providing biological control of insect pests, and bird damage to fruit in apple orchards in central Victoria and southern New South Wales. We found that: (1) excluding birds from branches of apple trees (via netting) resulted in 12.8% greater insect damage to fruit and reduced crop yields, therefore indicating that birds may contribute to controlling insect pests; (2) bird damage to apples on open branches was very low (1.9%), and only detected at two sites; and (3) the net outcome of bird activity, trading off the potential benefits of birds controlling insect pests vs. birds directly consuming fruit was positive, with the amount of damaged fruit reduced by 10.9%. The experiment using real and artificial codling moth larvae suggested that earwigs may significantly contribute to the predation of codling moth larvae in apple orchards, and this was more evident in less intense orchards. Real larvae were also predated on more often than plasticine ones. Hence, our combined results suggest that both insectivorous birds and certain predatory insects may together help control insect pests in apple orchards.

### Apple damage

The highest amount of insect damage to apples was recorded at Orchard 2, where almost 40% of apples on netted branches were damaged. Orchard 6 was the only orchard where trees received less damage on open branches than bird excluded ones, and overall insect damage was low compared with the other orchards. This was possibly due to low species richness of insects (ME Saunders & GW Luck, 2015, unpublished data) and birds, and relatively substantial pesticide use. It is likely that the lowest intensity orchards (Orchard 1 and Orchard 2), which were certified organic, had more insect damage because the growers did not spray any pesticides. It is important to note that the damage estimates in the higher intensity orchards reflect the combined influence of natural (biological) and chemical pest control, unlike the two organic orchards which use natural pest control alone.

Distance to unmanaged vegetation was not included in the analysis as it was moderately correlated with Orchard, however, native vegetation is important habitat for birds (*Bennett & Ford*, *1997*; *Tscharntke et al.*, *2008*; *Puckett et al.*, *2009*) and this is one possible explanation for why birds were providing greater biological control in the orchards closest to patches of unmanaged vegetation. The least intensively managed orchards had the highest insectivore richness (potential biological control providers), which also supports this conclusion. This was additionally supported by observations during bird surveys, with many insectivorous birds being observed at the orchard edges near unmanaged vegetation (e.g., yellow-faced honeyeaters (*Lichenostomus chrysops*) and grey fantails (*Rhipidura albiscapa*) darting to and from unmanaged vegetation and apple trees near the edges of the orchards).

Almost all (92.3%) of the 39 bird species observed were native. The greatest species richness occurred during the full bloom period, as did the greatest species richness of insectivores (however, these still occurred in fairly high numbers across the whole season). Between early fruiting and harvest, the apple trees in the two Harcourt orchards were covered with drape netting (excluding the open branches used in this experiment) and this appeared to provide a haven for many small bird species (e.g., striated thornbills (*Acanthiza lineata*) and yellow-faced honeyeaters) which could fit under gaps in the net,

while excluding most larger parrot species and birds of prey (e.g., black-shouldered kite (*Elanus axillaris*)).

Across the season (flowering, early fruiting and harvest) parrot species (crimson rosella (*Platycercus elegans*) and eastern rosella (*Platycercus eximius*)) were observed feeding on flower buds and fruit on the trees (possibly reducing fruit set) in all orchards except for the two in Shepparton (where no parrots were observed). These species were also observed consuming unharvested (waste) fruit on the trees and ground post-harvest, possibly reducing the spread of disease and increasing the transfer of nutrients and organic matter into the soil, which is particularly important in organic orchards (*Neeson*, *2008*; *Queensland Department of Agriculture and Fisheries*, *2015*). Satin bowerbirds were observed also feeding on apple flowers (detrimental activity) at Orchard 3 during flowering, and on fallen fruit (beneficial activity) in the same orchard during the early fruiting stage. Therefore, these bird species have the potential to be providing both costs and benefits to growers depending on the time of season. This supports work done by *Luck* (*2013*), who found that parrot species in almond orchards caused costs by eating the growing nuts before harvest, but then also provided benefits by cleaning up waste nuts after harvest.

The omnivorous silvereye (*Zosterops lateralis*) was observed eating nectar during flowering (possible detrimental activity of nectar robbery and possible beneficial activity of incidental pollination), gleaning insects from apple leaves and fruit across the whole season (potential beneficial activity of biological control), and eating unharvested or fallen fruit after harvest (potential beneficial activity of reducing the spread of disease and increasing the transfer of nutrients and organic matter into the soil). Such behaviour demonstrates the complex relationships between species activity and crop production, underscoring the importance of accounting for both the costs and benefits of the activity of multiple species across the entire growing season for individual crops in different contexts (see *Saunders et al.*, *2016*). Other omnivorous species, such as the Australian magpie (*Cracticus tibicen*) and European blackbird, were also identified in the orchards across the whole season and were only observed eating insects, some of which may have been apple pests.

## Larvae predation experiment

The larvae predation experiment found a significant difference in the prey type used, i.e., real larvae were preyed on more than plasticine larvae. This can possibly be explained by the high incidence of insect predation of larvae, particularly by earwigs. The larvae in this study were first attached to the study trees in the evening to allow birds to have the first morning of foraging uninterrupted (i.e., when they are most active); however, many of the real larvae were eaten by earwigs in the first night. This could have reduced the amount of larvae available for birds to prey on that first morning and therefore underestimated their biological control potential. Perhaps more importantly, earwigs seem to be largely overlooked as potential biological control providers of invertebrate pests in apple orchards. European earwigs are known to cause damage to many crop types (*Capinera*, *2001*), although they have also been shown to provide benefits such as controlling aphids (e.g., *Nicholas, Spooner-Hart & Vickers*, *2005*; *Romeu-Dalmau, Piñol & Espadaler*, *2012*). Native carnivorous earwigs (e.g., *Labidura truncata*) are better recognised

as potential predators (*Williams*, *2002*; *Atlas of Living Australia*, *2015*), but more research is needed into their potential as biological control agents, particularly in apple orchards.

Our study found that the use of plasticine larvae significantly under-estimates the amount of predation occurring in the orchard (i.e., they were attacked less than the real larvae). Other studies have found plasticine larvae to be effective in recording predation events (e.g., *Loiselie & Farji-Brener*, *2002*; *Koh & Menge*, *2006*; *Posa, Sodhi & Koh*, *2007*; *Howe, Lövei & Nachman*, *2009*; *Tvardikova & Novotny*, *2012*), however these studies did not directly compare their results with predation rates on real larvae. *Sam, Remmel & Molleman* (*2015*) considered this difference in prey type in a tropical forest system. They found no difference between real and artificial larvae; however, they found a significant difference in predation rates based on the type of artificial material used. Our study is therefore the first to consider the differences in predation rates on different types of prey items in fruit orchards. We suggest that while the use of plasticine prey models can help identify potential predators, and they may have some utility in recording differences in relative predation pressure among sites, they should not be used to infer actual predation levels on a given prey type. Our study also suggests that caution should be taken when using plasticine larvae to infer relative predation pressure. For this to be reliable, sites with high predation pressure for real larvae should also have high predation pressure for artificial larvae; however, this was not always the case in our study region.

Our study did not consider the potential for insectivorous bats to be providing biological control, although they can be important predators of crop insect pests (*Cleveland et al.*, *2006*; *López-Hoffman et al.*, *2014*; *Wanger et al.*, *2014*; *Brown, Braun de Torrez & McCracken*, *2015*). No bats were recorded on the motion sensor cameras, but several growers have observed insectivorous bats in their orchards, so this would be an important avenue for further investigation. Similarly, some growers commented on grey-headed flying foxes (*Pteropus poliocephalus*) causing serious damage to apple trees and fruit, but they were not observed during our study and there was no evidence of their damage (e.g., snapped branches and chewed fruit) (*Victorian Department of Natural Resources and Environment*, *2002*).

## The net value of bird activity

When trading off the amount of bird damage in the orchards (average of 1.9% of apples) with the amount of biological control they provided (i.e., reducing apples damaged by insects by 12.8%), it can be suggested that birds are providing an overall net benefit to orchard growers, reducing damage by an average of 10.9%. This value differed between orchards, with birds providing less biological control in the most intensively managed orchard (Orchard 6), and also causing more damage to apples than the insect control they provided in the least intensively managed orchard (Orchard 1) (i.e., a net outcome that was detrimental to growers). This highlights the importance of recognizing the spatio-temporal, management and ecological differences between orchards and understanding that there is no one-size-fits-all approach to sustainable management. Bird damage was only detected at two sites, Orchard 1 and Orchard 2, which were the two least intensely managed orchards. As insectivore species richness was also highest in these orchards, this suggests

that there is a point when the trade-off between encouraging insectivores (e.g., by planting native vegetation near orchards) may be out-weighed by attracting detrimental species (e.g., parrots). Though data are sparse, our results suggest that some active management of apple orchards is required to tip the balance in favour of a positive net outcome of bird activity for growers.

It is important to note that the bird damage recorded in our study was low and likely impacted by orchard management actions (e.g., drape netting). While some studies suggest that bird damage to apples is also low (e.g., *Long* (*1985*) who found a maximum of 1.75% damaged fruit per orchard), other studies have found bird damage in apples to be much higher, for example up to 18% in some varieties (*Grasswitz & Fimbres*, *2013*). These studies also found that the amount of damage depended on apple variety, with birds showing a preference for red-coloured, late season ripening fruit. Logistical reasons prevented us from controlling for apple variety, so all apple damage assessments were done at the same time in the season to account for temporal variation in bird activity. However, it is possible that apple variety could have impacted damage levels across orchards.

The amount of bird damage to crops can also vary between seasons and years (e.g., *Long*, *1985*; *Luck, Triplett & Spooner*, *2013*). In addition, there is a large amount of spatial variability in damage from large flocks of birds (e.g., cockatoo species) that descend somewhat randomly on localised areas in orchards (*Long*, *1985*). Therefore, it is possible that at other sites within the study orchards, or in future years, there may be more apple damage by birds. This is supported by personal observations of crimson rosellas and Australian king parrots (*Alisterus scapularis*) feeding on uncovered fruit near orchard edges at four of the orchards, and growers' observations of flocks of musk lorikeets (*Glossopsitta concinna*) feeding on fruit in previous years. Therefore, further work is needed to consider the cost-benefit trade-off of bird activity over larger spatial scales and longer time spans.

Our study begins to address the complex ecological interactions that occur between birds, invertebrates and apple crops. It highlights how birds can provide costs or benefits to growers depending on a range of contextual factors including time of season, location, and interactions with other fauna (i.e., invertebrates). This can better inform land managers about implementing strategies which promote the beneficial processes that are essential to the sustainability of agriculture and conservation alike (*Saunders et al.*, *2016*; *Peisley, Saunders & Luck*, *2015*), while reducing negative impacts on production.

## ACKNOWLEDGEMENTS

Many thanks to W Robinson at Charles Sturt University for assistance with the statistical analysis; D Williams and M Hossain at Agriculture Victoria for providing the live codling moth larvae; and A Peisley and two anonymous reviewers for comments on the manuscript.

### Funding

This research was funded by an Australian Research Council Discovery Grant DP140100709 awarded to GWL The funders had no role in study design, data collection and analysis, decision to publish, or preparation of the manuscript.

### Grant Disclosures

The following grant information was disclosed by the authors:
Australian Research Council: DP140100709.

### Competing Interests

The authors declare there are no competing interests.

### Author Contributions

- Rebecca K. Peisley, Manu E. Saunders and Gary W. Luck conceived and designed the experiments, performed the experiments, analyzed the data, contributed reagents/materials/analysis tools, wrote the paper, prepared figures and/or tables, reviewed drafts of the paper.

### Animal Ethics

The following information was supplied relating to ethical approvals (i.e., approving body and any reference numbers):

Charles Sturt University Animal Care and Ethics Committee; Approval number: 14/040.

### Data Availability

The raw data has been supplied as a Supplemental Dataset.

### Supplemental Information

Supplemental information for this article can be found online at http://dx.doi.org/10.7717/peerj.2179#supplemental-information.

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
