# Peer review of "Cost-benefit trade-offs of bird activity in apple orchards"

_PeerJ, doi:10.7717/peerj.2179_

## Round 0.1 · original submission · Major Revisions

As mentioned by two reviewers, this article adds to the existing knowledge of biological control of insect pests but they have also raised some concerns about the design of the experiments. In your response to these comments, please pay attention to the justification for the experimental design and use of baits in predation experiments.

Reviewer 1 ·

Basic reporting

The manuscript appears to follow the required format for the journal. Overall, the text of the manuscript was well written and very thorough in all sections of the paper.

Experimental design

The main concern I had with the paper was with the part of the experimental design involving the use of living and artificial codling moth larvae as part of an experiment to exam “larvae predation.” For me, this aspect of the study was very unrealistic. For example, on lines 204-209 in the manuscript, the authors describe taking nearly full-sized larvae (prior to pupation) and attaching them to a section of cardboard on a tree limb, either “close to apple clusters . . . where they naturally occur.” First, full-grown codling moth larvae DO NOT occur exposed in the spring and are not found near developing apple clusters. Such larvae have overwintered and are in protected sites getting ready to pupate and soon to emerge as adult moths. To place relatively large, non-moving larvae exposed on a light brown card (a mono-colored background) is in effect drawing the bird’s attention to the site. If that was not enough of a poor design, why did the authors not use a mottled, bark colored background on which to place the larvae? Similarly, and even more unnatural, was the placement of numerous larvae on a large cardboard tray that were placed in the trees. The only active, exposed stage of codling moth at this time of the season are the emerging adults that fly at dusk and during early evening hours where they lay eggs on apple foliage near developing fruit. As such, and in my opinion, any larvae predation data reported inn the manuscript are inaccurate to naturally occurring conditions and should have very limited value in assessing birds as effective biocontrol agents in apple orchards.

Validity of the findings

Despite my strong objections to the experimental design, the data the authors presented does seem to indicate, “that birds were contributing to the biological control on insect pests [i.e., codling moth?] in most orchards” (lines 258-259). At this point I have no data to disprove what the authors have reported, just that the larvae predation data should have very limited value. I just don’t see the logic in the experimental design. The authors in the Results section state, “several insectivorous bird species were observed within the orchards during the early fruit set and these could potentially be providing biological control of insect pests [i.e., codling moth?]” (lines 325-326). Again, I point out that most bird damage to apples, in which they are hunting for internal codling moth larvae, occur later in the season when the fruit is more ripe and softer for beak penetration. The larvae that are in the fruit at this time (“early fruit set”) are quite small and not that attractive to foraging birds (see author observations on lines 450-454).

In regards to the larvae predations study, I find the experimental design to be quite inappropriate because it is so unnatural from true field conditions and codling moth larvae behavior.

Additional comments

Due to my concerns about parts of the experimental design I am recommending that the manuscript receive “Major Revisions” before it is published. The authors need to do a better job in justifying the quite unrealistic design of the “Larvae Predation” study.

Reviewer 2 ·

Basic reporting

General

The role of birds on insect control in orchards certainly is an interesting topic. That it hasn't been documented to a greater extent in literature may have to do with the difficulty of collecting data. The current ms reminds us how difficult it is to design experiments and to finance the (wo)manpower needed to replicate them sufficiently many times and to collect experimental data with sufficient "resolution".

Orchards. Every single orchard is unique, selecting 1 representative orchard for different management practices isn't really sufficient to differentiate between orchard types, such as "organic" or "conventional".
This leads to the major question whether the statistical analysis is sufficient. Would it be meaningful to feed all measurements incl attributes into a matrix and submit this data to a multidimensional approach, followed by PCA, for example? The authors might be advised to collaborate with an expert. It isn't possible to derive general conclusions on orchard management type using data from single orchards, so it isn't meaningful to keep the orchard data apart and organize data presentation/evaluation by orchard.

Insect damage is classified as minor or major, which is questionable. A "pinprick hole" may stem from a neonate codling moth larvae, which destroys the apple, which isn't "minor" damage (line 163). Apples are rarely infested by more than one codling moth larva, and the damage doesn't become "major" only when this occurs. Also, there must be many different insect herbivores in these orchards, including surface feeders (such as leafrollers, aphids, etc.) - and it is a pity that the type of insect damage wasn't diffentiated, since birds would be expected to feed on different insects to a different extent.

Larval baits. Codling moth larvae are not visible to birds, they're either inside the apple or hidden (under bark) when hibernating (and they exit apples only at night). Birds will of course feed on visible/exposed larvae, which produces a measure of bird presence/feeding, but doesn't tell us anything about bird predation on naturally occurring codling moth larvae.

The insect fauna is fairly constant for a geographical region, despite orchard-to-orchard variation. In comparison, bird species are expected to vary much more according to orchard size, tree age, surrounding habitats etc. This is obviously an experimental difficulty.

The lack of resolution becomes obvious even with respect to details, such as the "intensity" of orchard management. Would it be an idea to evaluate the growers' logbooks and feed insecticide sprays (chemicals, application rate) into the data matrix, together with other orchard parameters? What does low vs high intensity mean - is that a measure of the author's perception? Tree age is another "orchard parameter", which is probably at least as important for bird feeding, trees 2 and 20 years of age are different worlds for insect foraging birds.

Last but not least, virtually throughout the paper refer to "birds" - without making all too many attempts to refine/resolve with respect to taxonomy or feeding habit. With a few exceptions, for ex. line 327: only 2 species feeding on larvae.
A main question arising is whether the conclusions are valid outside this very study area in Australia? Should the term "bird" [in abstract] be replaced with "all birds in one region in Australia" or should one instead attempt to specify bird species or even foraging behaviour, which would allow comparisons and predictions with/for other regions.

The data could well be good enough for publication, but the presentation and evaluation incl statistics needs a major revision.

Specific comments

Orchard abbreviations - these mean nothing to the reader, they should be replaced with either just numbers or abbreviations that indicate the management type. All location-specific data should be fed into data analysis.

line 27. rephrase? What else could be a positive contribution? Fertilization?
46. amount must be related to surface, otherwise meaningless.

47-49. Rephrase/delete - does this phrase provide information ?

Experimental design

see above

Validity of the findings

see above

Additional comments

see above

---

## Round 0.2 · accepted · Accept

Thank you for your response to the reviewers’ comments, which have been addressed clearly and adequately.

When you receive your article for proof-checking, please check the following minor edits:

Line 91: you can specify ‘south-eastern’ Australia

Line 311: Figure 7 is mentioned here but there is no Figure 7 in this version. Please check.

In supplementary information you have provided Figure S7 without referring in the main text. Please check.